# Review of the Biology, Distribution, and Management of the Invasive Fireweed (*Senecio madagascariensis* Poir)

**DOI:** 10.3390/plants11010107

**Published:** 2021-12-30

**Authors:** Kusinara Wijayabandara, Shane Campbell, Joseph Vitelli, Asad Shabbir, Steve Adkins

**Affiliations:** 1School of Agriculture and Food Science, Faculty of Science, University of Queensland, Gatton 4343, Australia; shane.campbell@uq.edu.au (S.C.); s.adkins@uq.edu.au (S.A.); 2Department of Agriculture and Fisheries, Brisbane 40000, Australia; Joseph.Vitelli@daf.qld.gov.au; 3Centre for Carbon, Water and Food, School of Life and Environmental Sciences, The University of Sydney, Camden 2006, Australia; asad.shabbir@sydney.edu.au

**Keywords:** CLIMEX, ecology, herbicides, impact, integrated control, pasture management

## Abstract

Whilst exotic invasive species are a major threat to natural and modified ecosystems around the world, management programs to reduce their impacts often fail due to a lack of information about their biology and how best to control them in various situations. This paper reviews the currently available information on the biology, distribution, and management options for the invasive weed *Senecio madagascariensis* Poir. (fireweed). In addition, we developed a model to predict the climatic suitability of this weed around the world based on the current climate. *Senecio madagascariensis* originates from southern Africa but it has been introduced to several other countries including Australia. Climatic suitability suggests that there are large areas around the world suitable for the weed’s growth where it is currently not present. The weed poses a major threat to livestock industries in these countries through its ability to reduce pasture production and poison animals. A range of control techniques have been used to try and manage *S. madagascariensis.* This paper highlights how a better understanding of the biology of *S*. *madagascariensis* can help determine the most effective treatments to impose and to further develop integrated management strategies. Besides using traditional approaches, the use of competitive pastures and more tolerant livestock (such as sheep and goats) are some of the other options recommended as part of an integrated approach. On-going research to identify host-specific biological control agents is also considered a priority.

## 1. Introduction

*Senecio madagascariensis* Poir. (fireweed), a native herbaceous plant from southern Africa [1] has been introduced to several countries including Australia, the United States of America (USA; Hawaii), Japan, Brazil, Argentina, Venezuela, Columbia, Uruguay, and Kenya [2]. *Senecio madagascariensis* plants contain pyrrolizidine alkaloids (PAs) and when eaten by some domestic livestock (particularly cattle and horses) it can lead to liver toxicity [3]. *Senecio madagascariensis* has an ability to spread into different habitats quickly through high seed production and multiple dispersal mechanisms, particularly wind. In some cases, effective *S. madagascariensis* management can be achieved using herbicides, but maintaining a healthy and competitive pasture and implementing multi-species grazing practices are also recommended for long-term control [4]. Nevertheless, the development of integrated management strategies that incorporate a range of options will be the most effective approach to reduce the impact of *S. madagascariensis* [5]. The aim of this paper is to evaluate the currently available information on the biology, distribution, and management of the invasive *S. madagascariensis* and to identify knowledge gaps that could be the catalyst for further research on this problematic weed.

## 2. Biology and Ecology of *S. madagascariensis*

### 2.1. Taxonomy

Botanical name *S.*
*madagascariensis* Poir. *Senecio madagascariensis* was first described by JLM Poiret in Madagascar in 1817 [6]. It belongs to the Tracheophytes, in the class Magnoliopsida, order Asterales, and family Asteraceae with synonyms of *Senecio bakeri* S. Elliot, *Senecio incognitus* Cabrera, and *Senecio ruderalis* Hary [7]. In South Africa, it is called *S. ruderalis* Harv. or *S. junodianus* O.Hoffm, whereas cytological studies in Australia have confirmed it as *S. madagascariensis* [8]. Its name is derived from Latin, senex meaning ‘old man’ referring to its pappus appearing like a white beard and *madagascariensis* meaning ‘from Madagascar’ [9].

The taxonomic position of Australian *S. madagascariensis* is undetermined. Initially, it was assumed to be part of the native (Australia) *Senecio pinnatifolius*; *S. pinnatifolium* (coastal groundsel) group which was made up of four clearly recognised sub-species in Australia viz. *S. pinnatifolius* sub spp., *alpinus*, *S*. *dissectifolius*, *S*. *lanceolatus*, and *S*. *maritimus* [10]. This is consistent with results from genetic analysis undertaken on northern QLD populations in Australia, which showed a close relationship with the *S. madagascariensis* complex from South Africa and only a moderate similarity to *S. madagascariensis* coming from Madagascar [11]. Furthermore, the species can vary in chromosome number with the coastal groundsels having a diploid chromosome number of 40 while *S. madagascariensis* has 2n = 20 for Australian [12] and Argentinian populations [13,14].

### 2.2. Common Name

*Senecio madagascariensis* is known by various common names including fireweed (Australia), Madagascar ragwort (Australia), and South African ragwort (Australia). Due to its ability to spread like a wildfire, the common name ‘fireweed’ is most used. Another possible reason for this name might be its bright yellow flowers appearing like a wildfire across the landscape. Other common names include, in Argentina “golden button” and “yellow flower of Mar del Plata” [15].

### 2.3. Species Description

*Senecio madagascariensis* is an erect, herbaceous, short-lived perennial plant ca. 10 to 50 cm tall (occasionally up to 70 cm). It forms a single stem or occasionally multiple stems which arise from a central crown at its base [15]. Its stems are multi branched, especially towards the top of the plant. Leaves are typically lanceolate with tips that are acute with denticulate margins [16] and sessile or sub-sessile [15]. Individual plants possess a branched tap root which grows 10 to 20 cm deep, with numerous fibrous roots [16].

*Senecio madagascariensis* possess heterogamous, radiate daisy-like flower heads that are canary yellow in colour, having a total ca. 120 small tubular disk florets and petal-like ray florets [9]. Florets are enclosed in involucral bracts (21) which are greenish in colour with brown or blackish tips [15]. A single plant can produce up to 200 flower heads (capitula) and ca. 30,000 seeds per year [17]. Fruit is small, ca. 1.5 to 2.2 mm long and up to 0.5 mm wide. Each fruit contains a single seed and a short hair-like structure, the pappus [18].

In Australia, *S. madagascariensis* can be confused with the coastal groundsels (*S. pinnatifolius*) as they both produce showy radiate inflorescences and grow to a similar height [19]. Moreover, both types are predominantly insect-pollinated and form fruit that are wind-dispersed [19]. The main distinguishing characters that separate *S. madagascariensis* from the costal groundsels are the number of involucre bracts at the base of the capitulum and the morphology of the seed [19]. *Senecio madagascariensis* has 20 to 21 involucre bracts/phyllaries (Sindel, 1989) while the coastal groundsels have ca. 13 cup-shaped involucre bracts but rarely 20 (each 3.0 to 7.5 mm long) with an apex that is weakly to strongly pigmented brown, black, or purple [20].

## 3. Economic Impacts of *S. madagascariensis*

The major economic impacts of *S. madagascariensis* are associated with a decrease in pasture productivity due to competition and poisoning of livestock [9]. Due to the production of toxins in *S. madagascariensis*, the livestock productivity losses in Australia have been estimated at $2 million annually [21].

According to the results of a national survey of landholders undertaken in Armidale and Dorrigo, NSW in 1985, almost 50% of respondents said that they spent more than 50 h every year controlling *S. madagascariensis* on their property, while about 40% said they spent ≥ $1000 on control actions [4]. In Hawaii, *S. madagascariensis* is distributed over 162,000 hectares of pasture [22], and US$11 million is spent annually trying to control it [23].

## 4. Geographic Distribution

### 4.1. Southern Africa—Native

*Senecio madagascariensis* is native to the southern parts of Africa, where it is present from coastal regions up to high altitude areas (i.e., 1500 m above sea level). It is considered native to Madagascar and the Mascarene Islands, to coastal Mozambique and KwaZulu-Natal and the Eastern and Western Cape Provinces of South Africa and Eswatini. In Madagascar, *S. madagascariensis* occurs in small remote populations at lower elevations in the southeast and in the semi-arid southwest of the island [11]. Besides its native range in Africa, *S. madagascariensis* has also been recorded as an introduced species in the highlands of Kenya [24].

### 4.2. Australia—Introduced

*Senecio madagascariensis* was thought to have reached Australia in the ships trading between Europe and Australia that had passed through the Cape of South Africa [8]. In 1918, *S. madagascariensis* was first identified close to Raymond Terrace in the Hunter Region, of central NSW. By the late 1960s, it was found almost 650 km further south along the shoreline in the Bega Valley, of south-costal NSW, possibly arriving there as a fodder contaminant from the north-coast of NSW. *Senecio madagascariensis* had become a pasture threat by about 1983, despite being a very dry year with little fodder having been produced [25]. Large populations of *S. madagascariensis* continued to establish and spread rapidly in the mid-1980s throughout the pastures of coastal NSW as well as into Southeast Queensland (SEQ) [26]. According to Sindel [27] isolated plants were also encountered inland, especially at Dubbo on the Central Western Plains of NSW, inside the limits of the Western Plains Zoo.

Moreover, in southern Queensland, *S. madagascariensis* has spread northward from Brisbane, with isolated infestations occurring near Caboolture, Cooroy, Belli Park, Maleny, Yandina, Pelican Waters, and as far north as Malanda. There is also an herbarium record from near Longreach, although the specimen appears to have been collected near the roadside and it is not clear if it was an isolated plant or if an infestation was present. A prediction in view of climate change and land utilization, suggests that *S. madagascariensis* could become a serious problem as far north as Rockhampton [28].

In 2007, the weed was reported close to Rockhampton and on the Atherton Tablelands in far north Queensland (QLD), in accordance with the expectations of Sindel and Michael [24]. In SEQ, the most widespread incursions of *S. madagascariensis* have been identified in Logan City, Gold Coast City, and Scenic Rim Regional Council areas. Although locally abundant, it is generally less common in Brisbane City, Redland City, Ipswich City, and Moreton Bay Regional Council jurisdictions [29].

### 4.3. Hawaii—Introduced

*Senecio madagascariensis* was first discovered in the archipelago of Hawaii in the 1980s where it had established in pastures near Waimea and Kona on the Big Island [22]. Since then, it has spread to pastureland in the north eastern and western sides of the island and increased towards the southern areas [21]. The naturalised *S. madagascariensis* populations are found from sea level up to 1600 m in Maui [21] and Kaua’I [30].

*Senecio madagascariensis* plants in Hawaii, according to molecular investigation, are like those from South Africa more so than those from Madagascar or Eswatini [21]. These populations are thought to have arrived in Hawaii as contaminants of *Axonopus fissifolius* (Raddi) Kuhlm (carpet grass) seed lots sent from Australia. A detailed comparative study was undertaken on the alkaloid content of *S. madagascariensis* plant material collected from a single location in Australia (northern NSW) and multiple locations in Hawaii [3]. In total, 13 pyrrolizidine alkaloids (PAs) were identified and included: senecivernine, senecionine, integerrimine, senkirkine, mucronatinine, retrorsine, usaramine, otosenine, acetylsenkirkine, desacetyldoronine, florosenine, and doronine. Interestingly, while there was large variation in total PA content amongst individual plants within locations (217 to 1990 μg g^−1^ on a dry weight basis); overall, the plant material collected from the Hawaiian Islands was found to be identical in pyrrolizidine alkaloids (Pas) content to the Australian collection. It was suggested that this was further evidence that *S. madagascariensis* in Hawaii may have originated from Australia [3].

### 4.4. Japan—Introduced

In 1976 *S. madagascariensis* was first identified in Naruto City in the Eastern Shikoku region of Japan. Gradually, it has spread widely in southern parts of the country. At present, *S. madagascariensis* covers a large portion of the southwestern part of Japan. Moreover, it has been observed on the Pacific coast and on the seacoast of Seto Island. According to these previous studies, *S. madagascariensis* has been confined to the warmer southwesten regions [2], including Tokyo where there is 1000 to 1700 mm of precipitation annually [8]. The pathway of its arrival into Japan is unknown.

### 4.5. South America—Introduced

Several countries in South America have recorded the presence of *S. madagascariensis*; Argentina, Brazil, Columbia, and Uruguay. In Argentina and Brazil, the first specimens of *S. madagascariensis* were collected in 1940 and 1995, respectively [31]. In Argentina it was found around cities and on roadsides in Buenos Aires province in the 1940s, but over about a 30-year period it spread into nearby provinces and is now widely distributed in northern and central Argentina [32]. In Brazil, it spread quickly throughout the Pampas region and is now widely distributed in southeastern Brazil [31,32]. In the 1980s, *S. madagascariensis* was recorded from Colombia for the first time [33]. It has invaded the cool moist Colombian highlands near Bogota [9]. In Uruguay, *S. madagascariensis* has been regarded as a serious threat to producers since its detection in late 1990s in grazing pastures near the western cities of La Concordia and Dolores in the region of Soriano and near the southwestern region of Colonia [34]. By 2010, farmers from the open eastern rangelands of Uruguay bordering Brazil were recording significant increases of seneciosis (intense acute or chronic necrosis of liver) in their grazing livestock [35,36].

## 5. Potential Spread and Future Distribution of *S. madagascariensis*

To assess the climatic suitability of *S*. *madagascariensis* around the world, a process-based model CLIMEX (version 3) has been used [37]. The CLIMEX model compares the response of a species to long term averages of climate for different locations. A series of growth and stress indices are inferred based on climates of known occurrences of a species. The annual growth index, GI, describes species response to temperature and soil moisture while stress indices (cold, wet, hot, dry, cold-wet, cold-dry, hot-wet, and hot-dry) exclude it from unfavourable locations. By combining these growth and stress indices, an Ecoclimatic Index (EI), ranging from 1–100 is calculated, which provide an overall measure of the suitability of a given location.

We used the ‘Compare Locations’ model in CLIMEX which was parametrised by using distribution data of *S. madagascariensis* from its native range in South Africa and Madagascar, and its introduced range, such as Japan. Growth and stress parameters of *S*. *madagascariensis* were fitted using the methodology described in [37,38,39]. A modified CLIMEX temperate template was used as a starting point. The model fitting strategy started with fitting of stresses based on known occurrences in native range in Southern Africa (South Africa, Swaziland, Mozambique, Madagascar; n = 242) and introduced range in east Africa (Kenya; n = 13) and Southeast Asia (Japan; n = 66). The stress parameters were fitted iteratively by changing the values until a best fit is achieved. To exclude the *S*. *madagascariensis* suitability in tropical regions of the world, a hot-wet stress (HDS) was used. Similarly, the cold stress (THCS) value was increased to constrain weed’s suitability from very cold regions of the world (Table 1).

The growth functions (temperature and moisture indices) of the model were fitted based on *S. madagascariensis* distribution in native range and introduced range, as well as reported experimental data [9]. To validate the model, the climatic suitability projection was matched with known occurrences of *S. madagascariensis* in Australia (n = 9283) and South America (n = 122). All occurrences in Australia and South America fall within projection, providing confidence that this model is reasonable representation of climatic suitability of *S. madagascariensis*.

The CLIMEX model, run under the current climate scenario, satisfies the present geographic range of *S. madagascariensis* around the world (Figure 1). All occurrences of *S. madagascariensis* matched well within the projection created by this model. For South America, the model suggests that most of the southern parts of Brazil, the whole of Uruguay and northeastern Argentina is highly suitable for *S. madagascariensis* (Figure 1). Most parts of Andes Mountain range in Bolivia, Peru, Ecuador, and Columbia are also suitable. In North America, southeastern States (Florida, Louisiana, Texas, Alabama, Georgia, Mississippi) and whole of east coast of USA, southern Mexico, and most of the Caribbean Islands are projected to be highly suitable for *S. madagascariensis*. One other highly suitable region, presently outside the range of *S. madagascariensis*, was in southeast Asia with southern and southwest parts of mainland China, southern Japan, and northern Vietnam being suitable with EI values of between 21 to 40 (Figure 1). Most of eastern and southern Africa, as well as northern most parts of African continent were predicted to carry a similar climate to that of the native range of the weed. According to the potential distribution model, most parts of western Europe including, Portugal, Spain, Italy Belgium, Hungary, France, Germany and the United Kingdom are all highly suitable with EI values of 30 to 60 (Figure 1). The model has predicted that southeastern Australia is climatically highly suitable for fireweed compared to northern Australia. This was further confirmed by concentration of most of the species’ occurrences within this region. Inland QLD and NSW and northern parts of the Northern Territory (NT) and inland Western Australia (WA) have EI values in the range of 1 to 20. Furthermore, many parts of Victoria and Tasmania are suitable for *S. madagascariensis* to grow. Although *S. madagascariensis* has so far not been reported from New Zealand, our model predicts that most parts of both North and South Islands are climatically suitable for *S. madagascariensis.* However, CLIMEX projections are very coarse and do not include the regional microclimate variations, biotic factors (competition, natural enemies, allelopathy, etc.) and land use practices that can also affect a locations suitability for invasion.

Sindel and Michael [24] using BIOCLIM, suggested the potential range of *S. madagascariensis* in QLD was from the southern border, along the southern coast and north to about Gympie. However, Csurhes and Navie [29] using CLIMEX, suggested the potential range could go further, north into coastal central Queensland and into higher elevation areas including the Atherton Tableland in northern Queensland.

### 5.1. Preferred Habitat

*Senecio madagascariensis* can develop a different growth habit, with different leaf shapes on different soil types and in different habitats [11]. *Senecio madagascariensis* is an opportunistic weed and it can adapt and spread into new areas rapidly [8]. The preferred habitat of *S. madagascariensis* in the invaded range includes roadsides, livestock feeding areas, and areas around dams and other wet areas [18]. It can also be found in improved grasslands, in meadows and along riverbanks [40]. *Senecio madagascariensis* prefers well-drained, non-compact, high fertility soil, but it can grow on a wide range of soils including sands and limed soil [8,9].

### 5.2. Climatic Requirements

*Senecio madagascariensis* prefers a humid, maritime, and sub-tropical climate. It is found at similar latitudes on the eastern coast of the three continents of the southern hemisphere with an annual rainfall of between 500 to 1000 mm [9]. An annual mean temperature of 12.4 to 20.1 °C favours the establishment of *S. madagascariensis* in most locations where it has invaded [9]. However, young seedlings are sensitive to frost while older plants show tolerance [21], but frost can reduce their vigour [24]. This may lead to its absence in areas with a high frost rate [21]. *Senecio madagascariensis* can grow at altitudes of up to 2800 m above sea level in the tropical environments of Kenya and Columbia suggesting that the warmer altitudinal temperatures in these countries may allow it to grow at higher altitudes than would normally be seen in more temperate countries [24].

Based on an empirical modelling approach, Le Roux et al. determined the most climatically suitable areas for *S. madagascariensis* in Hawaii [41]. They concluded that they were low elevation and arid areas on the windward sides of all islands, with minimum and maximum annual temperatures between 12 to 18 °C and 21.2 to 26.5 °C respectively, elevations between sea level and 1000 m, solar radiation levels between 350 to 400 calories m^−2^ day^−1^, and annual precipitation between 178 to 376 mm year^−1^. This is consistent with other studies, although the estimation for precipitation is substantially lower. A potential limitation of this study is that the model developed is only based on climate data taken from positive locations on the five major islands of Hawaii. In addition, the LeRoux’s projection (41) did not completely exclude the higher elevation and humid areas as possible invasion sites, as they were predicted to be reasonably suitable.

### 5.3. Growth and Development

*Senecio madagascariensis* is a short-lived perennial, however it often grows as an annual. Most plants finish their life cycle at the end of their first year [9]. However, a few plants will remain, continue to grow, and reproduce vigorously throughout their second year and therefore under these circumstances it is regarded to be a perennial. It can exhibit high plasticity with the ability to change its life cycle depending upon local conditions [9].

### 5.4. Reproduction and Seed Dynamics

*Senecio madagascariensis* reproduces primarily by seeds (i.e., achenes), although vegetative reproduction has been observed under certain conditions [8,42]. When its stems are trampled and contact moist soil, roots and shoots can sprout from the stem’s nodes, resulting in new, self-supporting plants [42].

Flowering can take place throughout the year in some countries [23] however, in Australia, it is typically from late autumn (May) through to mid-summer (January). If conditions are favourable, some plants flower until late summer (February). Generally, plants flower 42 to 70 days after seedling emergence. Flowers are pollinated by insects such as European honeybees and hover flies [9]. Generally, the plants undergo senescence and die after flowering as this is an important part of the life cycle.

Shed seed can germinate as soon as it is dispersed [16]. Although, seed can germinate throughout the year, in Australia the highest peak in germination occurs from March to June [15]. Seeds germinate within a temperature range of 15 to 27 °C at the soil surface [18], and maximum germination occurs between 20 to 25 °C and germination ceases at 35 °C or above [9]. Germination is also highest in the presence of light compared to dark conditions [12]. Under normal conditions, *S. madagascariensis* seed dormancy is negligible and high temperatures induce seed dormancy [43]. Despite having relatively high germinability, viable seedbanks are thought to persist for between 3 to 5 years [18] and potentially up to 10 years under certain conditions [9].

### 5.5. Seed Dispersal

Human aided dispersal has been responsible for the introduction of *S. madagascariensis* into at least eight countries outside of its native range. Once in a new environment, the seeds can be dispersed from local populations through multiple mechanisms including wind, by attaching to animals and vehicles, or in contaminated agricultural produce [29]. Wind has been attributed to the rapid spread and expansion of *S. madagascariensis* in parts of several countries, including Australia, Argentina, Brazil, and the USA (i.e., Hawaii) [32,40,44]. It was hypothesised that the spread of *S. madagascariensis* by wind might have been enhanced through the rapid evolution of superior dispersal traits at the invasion front (i.e., range edge populations) [40], as has been reported for *S. inaequidens* in Europe [45]. However, after comparing key features of the propagules of *S. madagascariensis* plants (i.e., pappus size, achene size, and the ratio of pappus size to achene size) growing on the edge and from within the established range, no differences in dispersal potential were found, which is advantageous from a management perspective. Bartle et al. [40] recorded a similar response for South American populations. However, they did find that wind dispersal of *S. madagascariensis* was strongly affected by adaptation to prevailing geographic conditions. In low altitude areas of Argentina, plants had adapted to produce smaller seeds that could be dispersed further by wind. In higher altitude areas, larger seeds were produced, which suggested that seedling establishment was favoured over long-distance dispersal [44]. In-addition to wind dispersal, it can be dispersed by hay, grain, clothing, vehicles, livestock, birds, and other animals [9]. However, it is unknown whether *S. madagascariensis* seeds can go through the digestive tract of cattle, sheep, or birds and then germinate [9].

### 5.6. Toxicity

Among the 1200-worldwide species of *Senecio*, ca. 25 species (including *S. brasiliensis, S. heterotrichius*, *S. cisplatinus*, *S. selloi*, and *S. oxyphyllus*) are toxic to certain livestock, such as horses (*Equus caballus* L.) and cattle (*Bos taurus* L., *Bos indicus* Brahman) [3]. According to various studies, some *Senecio* species can be toxic to humans. For example, if milk products containing PAs from *S. madagascariensis* are consumed [46].

Due to the presence of pyrrolizidine alkaloids (PAs) in the leaves, *S. madagascariensis* is highly toxic to certain domesticated animals; that is, it can cause hepatopathy, decrease the development of young animals, and in some cases cause mortality when ingested [2]. According to Sheppard et al. the PAs from *S. madagascariensis* could cause chronic liver damage and fatality of horses [47]. Weeks or months after feeding on the plant has ended, the symptoms and death of animals can still occur. Although moving stock to areas free from *S. madagascariensis* can prevent further progress of the disease, ill health, and poor growth may continue [9].

In countries such as Brazil, Columbia, and Uruguay, where *S. madagascariensis* is spreading into new areas, there is growing concern that it is contributing to an increase in PA poisoning [33,35,48].

## 6. Management of *S. madagascariensis*

Several management techniques are used to control *S. madagascariensis*. These include cultural, physical, chemical, and biological methods, or a combination of these methods.

### 6.1. Legislation

In Australia, *S. madagascariensis* is a declared weed under the relevant legislation of all states and territories. However, the level of declaration and associated requirements varies greatly depending on the perceived level of risk. *Senecio madagascariensis* was added to the Hawaiian State Noxious Weed List by their Department of Agriculture in 1992 [30]. In Japan *S. madagascariensis* has been declared an Invasive Alien Species under the Invasive Alien Species Act (which restricts importation, moving, or growing of species within Japan) [2].

### 6.2. Physical Control

*Senecio madagascariensis* seeds are mostly wind dispersed and therefore the physical hand pulling of plants must be completed before the seed is formed if it is to be effective. The pulled plants should then be burnt or deep buried to prevent plants re-growing and producing further seeds. This technique is only practical for isolated plants or small patches and not for large infestations, as it is time consuming and labour intensive [4,18,42].

In relatively flat and accessible areas, mowing and cutting style equipment (e.g., slashers, mulchers) can be used to weaken *S. madagascariensis* plants and help prevent them from reaching reproductive maturity [42]. To be effective, this technique needs to be repeated as few *S. madagascariensis* plants will be killed from a single operation [4,42]. It will also be most effective on smaller plants and if applied when the pasture is growing, enabling maximum competition to be imposed to restrict *S. madagascariensis* regrowth [18].

In the central coast region of NSW (Australia), slashing and mulching of paspalum (*Paspalum dilatatum* Poir.) and kikuyu (*Pennisetum clandestinum* Hochst. ex Chiov.) pastures infested with *S. madagascariensis* from mid-September onwards is an effective strategy for its management [16]. However, mulched *S. madagascariensis* can wilt and become more attractive to stock feeding on it and contains a greater concentration of PAs. Thus, after slashing or mulching, *S. madagascariensis* infested paddocks should not be grazed for at least 2 weeks [9]. Slashing can also have other downsides such as facilitating further spread of *S. madagascariensis* if plants are reproductive, and it may only delay flowering until later in the season and promote regrowth of plants in the following season [18].

### 6.3. Chemical Control

In terms of chemicals, 2,4-D formulations [16,22], dicamba, glyphosate, MCPA, tebuthiuron, triclopyr [22], bromoxynil, fluroxypyr/aminopyralid, metsulfuron-methyl, and triclopyr/picloram/aminopyralid [18,28] are some herbicides that have been found to be effective on one or more growth stages of *S. madagascariensis*. However, which chemicals and rates that can be legally applied, may vary between countries and even between jurisdictions within countries.

The selective herbicide bromoxynil can be very effective on young plants, but mature plants are more tolerant. Being a contact herbicide, only those parts of a plant that come directly in contact with the herbicide are killed and the plant will often regrow from unaffected parts. Significant seedling recruitment after spraying is also often observed. It has been reported [28,49] that bromoxynil at 3 L ha^−1^ was unsuccessful at controlling mature *S. madagascariensis*, with substantial regrowth recorded five months after spraying.

Glyphosate, a nonselective systemic herbicide proved equally as effective as bromoxynil in trials undertaken in Argentina [50]. However, as with bromoxynil, re-infestation will occur afterwards. Additionally, as glyphosate is nonselective it should only be applied where non-target damage can be tolerated or using target specific equipment such as wipe-on applicators [22].

In Australia, 2,4-D amine (3.2 kg ha^−l^) and 2,4-D sodium salt (2 to 4 kg ha^−l^) have been reported to give positive results after spraying, without harming the neighbouring pasture species, such as blue couch (*Digitaria didactyla* Wild), cogon grass (*Imperata cylindrica* (L.) Beauv), and white clover (*Trifolium repens* L.) [49]. Similarly, Ref. [22] suggested that in Hawaiian pastures where forage legumes are mixed with grasses, the amine salt formulation of 2,4-D would be preferable because of its mild effect on legumes. In contrast, metsulfuron-methyl at 40 to 80 g ha^−1^ provided effective control of *S. madagascariensis* in an Australian study, but it was severely damaging to any legumes (such as *T. repens*) present within the treated pasture [49].

Whilst the efficacy of several herbicides on plant mortality is known, there is minimal information on their effect on seeds located on plants at the time of their application. This is important, as landholders often spray mature plants, and it would be advantageous if the herbicides used not only killed the plants but also kill any seeds located on them at that time. It has been suggested that herbicides do not generally kill *S. madagascariensis* seeds if applied after flowering [18], but some success has been reported using certain herbicides on *C. odorata* (L.) R.M.King & H.Rob. (Siam weed) another Asteraceae species. According to one study metsulfuron-methyl was effective on immature and intermediate seed maturity stages but not mature seeds [51], whilst fluroxypyr was highly effective at causing mortality of immature and intermediate seeds and rendered a proportion of mature seeds non-viable after the plants were sprayed.

Even using the most effective herbicide, a single application will not suppress *S. madagascariensis* permanently [22] and follow up control will be necessary. Consequently, management strategies that rely solely on herbicide applications can become expensive if large areas must be controlled. Chemical control of *S. madagascariensis* in the Hawaiian archipelago, if undertaken, would cost USD $11 million year^−1^ [11], making it an uneconomical proposition [21].

### 6.4. Biological Control—Competitive Pastures

In an effective *S. madagascariensis* management program, maintaining a vigorous, competitive pasture through the autumn and winter months is an important step in providing best management practice [9]. The summer-growing and high fodder yielding perennial pasture species such as setaria (*Setaria sphacelata* Schum.), kikuyu (*Pennisetum clandestinum* Hochst. ex Chiov.), paspalum (*Paspalum dilatatum* Poir.), and Rhodes grass (*Chloris gayana* Kunth.) are also likely to be suppressive throughout the winter months. These predominantly summer-growing pasture species can be grown through the late summer and autumn months, to provide good groundcover, which then helps to prevent *S. madagascariensis* seedling establishment in the autumn and winter months [52]. However, according to a study undertaken in South Africa, *Eragrostis curvula* Schrad., *Cynodon dactylon* (L.) Pers., *Dactylis glomerata* L., *Festuca arundinacea* Schreb., *Pennisetum clandestinum* Hochst. ex Chiov., and *Themeda triandra* Forssk. grasses were all unable to suppress *S. madagascarinsis* due to its highly competitive ability [53].

### 6.5. Biological Control—Insects and Pathogens

Although a biocontrol program to control *S. madagascariensis* first commenced in Australia in 1987, only two insects were tested, and neither were released. *Aecidium* sp. is a rust fungus that was imported into quarantine in Australia for detailed studies. This rust was like a naturally occurring Australian isolate of the orange rust *Puccinia lagenophorae* Cooke [54] that is found on the family Asteraceae. The South African rust was less damaging than the Australian isolates of *P. lagenophorae.* These results imply that their introduction would be doubtful to Australia due to its low virulence, which would be expected to translate to poor control of *S. madagascariensis* [54].

In 2013, *Secusio extensa* (Erebidae: Arctiinae) was introduced to control *S. madagascariensis* in Hawaii. However, since release, this agent has undergone population outbreaks on a related alternative host, Cape ivy (*Delairea odorata*, Asteraceae: Senecioneae) which is another noxious weed in Hawaii. Given that this herbivore significantly preferred Cape ivy over *S. madagascariensis* for its oviposition and larval feeding, it has not proven to be an effective biological control agent in Hawaii [55]. Generalist herbivores (e.g., grasshoppers, weevils, thrips, aphids, scale insects, whiteflies, and vermin) that feed upon *S. madagascariensis* in Hawaii have been also shown to have an insignificant effect on the plant [11].

In the KwaZulu Natal Province of South Africa, 12 natural enemies have been recently identified in initial surveys feeding on five populations of *S. madagascariensis*. They comprise of three stem borers, four flower feeders, two sap suckers, and three plant pathogens [17]. Recently, four potential agents were prioritised for host-range assessments and these studies are currently in progress. The agents being tested are the capitulum-feeding *Homoeosoma stenotea* Hampson (Lepidoptera: Pyralidae), the stem-boring *Gasteroclisus tricostalis* (Thunberg) (Coleoptera: Curculionidae), and *Metamesia elegans* (Walsingham) (Lepidoptera: Tortricidae) and the root-feeding *Longitarsus basutoensis* Bechyné (Coleoptera: Chrysomelidae: Alticinae) [56].

### 6.6. Livestock Grazing

Using cattle as a control strategy for *S. madagascariensis* tends to be unsuccessful, as the reduced competition and improved light conditions that occur once the pasture is grazed down allows new seedlings to grow faster. Cattle also tend to avoid it in heavily grazed paddocks as it is more easily distinguished amongst the grass pasture [9]. In contrast, grazing with sheep or goats is considered an effective method to control *S. madagascariensis*. According to Watson et al. sheep and goats are about 20 times more tolerant to *S. madagascariensis* poisoning than horses and cattle [16]. Sheep might ingest and suppress *Senecio* spp. toxin due to their ability to undergo hepatic detoxification. This is related to their ruminal flora populations that can reduce the probability of poisoning [57].

In southern Brazil, high densities of *S. brasiliensis* and *S. madagascariensis* were controlled by using 16 sheep (3.0 stock units ha^−1^) in a 5.5 ha experimental area [57]. A total of 28,629 plants of *S. brasiliensis* (10,122) and *S. madagascariensis* (18,507) were almost eliminated within a 2-year period. Before and after the experimental period, liver biopsies of sheep and cattle were examined but did not reveal any sign of seneciosis [57].

The effectiveness of using herbivores such as sheep and goats will depend a lot on the number of propagules available for recruitment to occur. In a Hawaiian study that investigated the effects of feral goats and sheep on *S. madagascariensis* in a natural environment (dry forest), Questad et al. found that in areas where there were high numbers of propagules, *S. madagascariensis* plants were smaller due to grazing, but there was largescale seedling recruitment [58]. In contrast, if there were few propagules at the start, recruitment and overall biomass of *S. madagascariensis* was greatly reduced.

Using livestock mediated herbivory, as part of a control strategy for *S. madagascariensis* is most applicable for modified environments such as pastures. In natural environments, careful consideration would need to be given as there is a higher risk of non-target damage occurring, but in some situations, it may still be applicable [58].

### 6.7. Integrated Management

Integrated weed management (IWM) is the use of a combination of methods, including cultural, physical, biological, and chemical approaches [59]. For *S. madagascariensis*, some trials to determine the effects of individual treatments have been undertaken, but testing of combinations of treatments is lacking. Despite this, Sindel and Coleman suggest that an effective IWM approach for *S. madagascariensis* control should include the use of perennial competitive pasture species, hand weeding (for isolated plants or small patches), the maintenance of good farm sanitation practices and the use of herbicides to control the weed [18]. Furthermore, for long-term control of this weed, reduced grazing within a competitive pasture may need to be used by landholders [18]. The integration of livestock that are more tolerant (e.g., sheep and goats) of *S. madagascariensis* should also be considered given the success obtained in reducing its density in some situations [57].

In Hawaii, Thorne et al. recommended that for the successful control of *S. madagascariensis*, federal, state, county, and private land managers need to develop IWM plans [42]. They further suggested that these plans be part of any management program for all land units, and that the key objectives should include preventive measures where the weed is not present, control measures where the weed is already established, and protocols for taking prompt action when *S. madagascariensis* first appears in an area.

To achieve this, an adaptive management approach was developed based on six key steps: establishing goals, setting management priorities, identifying appropriate methods, developing, and implementing an integrated weed management plan—monitoring results, modifying priorities, and improving the management plan. In terms of control options, they highlighted that they should, (a) not contribute to the spread of the weed, (b) be applied at the most effective time (i.e., point in the life cycle when it is most vulnerable), (c) minimise the risk to human health and potential damage to the general environment (e.g., non-target species), and (d) be cost effective. Key benefits and limitations associated with various options (i.e., pulling, mowing, cutting, cultural controls, livestock grazing, biological control agents, herbicides, burning) are also outlined to help landowners/managers to make informed decisions [42]. A working model is now proposed for the improved management of *S. madagascariensis* (Figure 2), however further refinement of the model is necessary.

## 7. Conclusions and Future Directions

Since *S. madagascariensis* contains PAs, that cause liver damage, certain domesticated animals (cattle and horses) can be poisoned after consuming this plant. It has been found to be a fast growing annual/short lived perennial plant that reaches reproductive maturity quickly and produces large quantities of viable seed for future recruitment events. The predicted distribution suggests that many regions across the world have a climate suitable for the growth of *S. madagascariensis*. This indicates that the weed is likely to expand its range in future, if not controlled. Although there are many management techniques available including legislation, cultural, physical, chemical, and biological (insects and suppressive plants), individually they have little effect on *S. madagascariensis* management. An IWM program incorporating a combination of techniques will generally be much more effective. However, for those landholders who do not have *S. madagascariensis* or are in the early stages of invasion, early detection is even more important to prevent its establishment in the field.

To build on the currently available information, further research is needed to improve management options for this problematic weed. In terms of ecology, seed dynamics appears to be a key factor influencing the spread and persistence of *S. madagascariensis*. Further research into the environmental conditions affecting germination, viability, and persistence would be advantageous, particularly if changing climate scenarios are included. From a control perspective, a range of herbicides are currently available to kill small to large *S. madagascariensis* plants. However, their effect on the germination and viability of seeds located on plants at the time of spraying has not been fully evaluated and warrants investigation. This is important as plants are often reproductive at the time of spraying and if a herbicide can not only kill the plant but also the seeds, subsequent seedling recruitment will be reduced. Control efforts will also benefit from the release of effective biological control agents that adversely affect key life stages of *S. madagascariensis*. Limited success has been achieved to date, but current efforts should continue in the search for host specific options. In the interim, improved pasture management and testing of suppressive pasture plants should be explored further as part of longer-term strategies to manage well established infestations. How these pastures are grazed, particularly in terms of the type of animals, is an area that could also be the focus of additional research. Animals such as sheep and goats are more tolerant of *S. madagascariensis* than cattle and utilising them in traditional cattle grazing operations could be beneficial. Finally, larger integrated management trials should be undertaken over different seasons and at several locations to encompass a range of pasture types, climatic conditions, and soil types. In doing so, detailed cost/benefit analysis data could be collected to enable landholders to make informed decisions about the most appropriate options for their situation.

## Figures and Tables

**Figure 1 plants-11-00107-f001:**
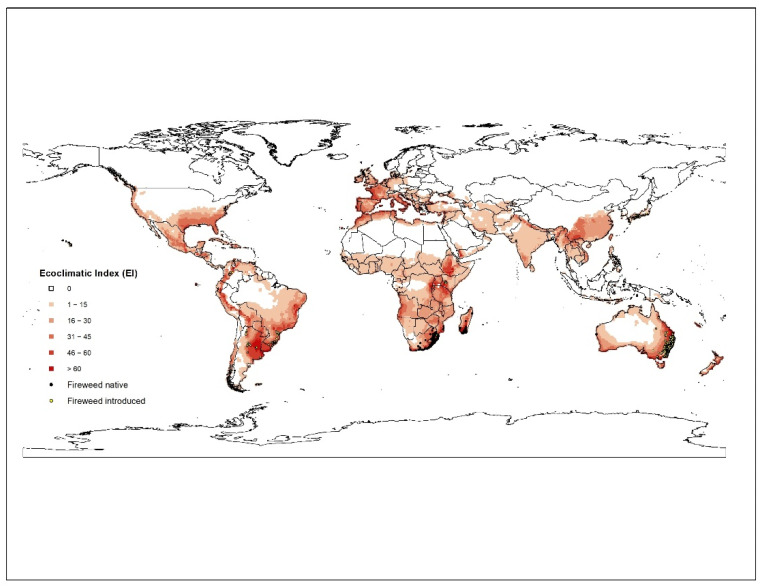
Current distribution (native—green dots; introduced—blue dots) and the climatic suitability for fireweed under the current climate modelled using CLIMEX.

**Figure 2 plants-11-00107-f002:**
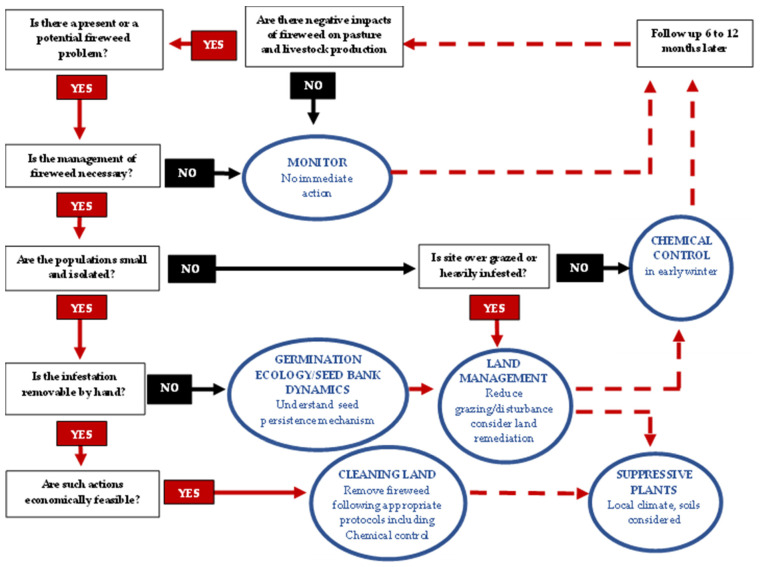
Conceptual decision diagram for proposing approaches to the integrated man-agement of fireweed incorporating managerial, chemical, and manual control.

**Table 1 plants-11-00107-t001:** CLIMEX parameter values used for modelling the potential distribution of *S. madagascariensis*.

Parameter Group	Parameter	Value	Units
Temperature index	DV0 = limiting low temperature	5	°C
	DV1 = lower optimum temperature	18	°C
	DV2 = upper optimum temperature	24	°C
	DV3 = limiting high temperature	35	°C
Moisture index	SM0 = limiting low soil moisture	0.1	mm
	SM1 = lower optimum soil moisture	0.2	mm
	SM2 = upper optimum soil moisture	1	mm
	SM3 = limiting high soil moisture	1.5	mm
Cold stress	TTCS = temperature threshold	2.5	°C
	THCS = stress accumulation	−0.0001	week^−1^
	DTCS = degree-day threshold	0	day °C
	DHCS = degree-day stress rate	0	week^−1^
Heat stress	TTHS = temperature threshold	35	°C
	THHS = stress accumulation rate	0.001	week^−1^
Dry stress	SMDS = wet stress threshold	0.1	week^−1^
	HDS = stress accumulation rate	−0.0001	
Wet stress	SMWS = wet stress threshold	2	week^−1^
	HWS = stress accumulation rate	0.001	
Hot-wet stress	PHW = stress accumulation rate	0.001	week^−1^

## Data Availability

Not applicable.

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
