# Peer review of "Review of the Biology, Distribution, and Management of the Invasive Fireweed (Senecio madagascariensis Poir)"

_plants, 2021, doi:10.3390/plants11010107_

Round 1

Reviewer 1 Report

I would like to report my review on the manuscript, entitled as Review of the biology, distribution, and management of the invasive fireweed (Senecio madagascariensis Poir).

After I read through its manuscript, I found it is well-written and informative document about a single invasive plant. It is very descriptive at the beginning and its extensiveness is acceptable given its subject.

I think the most important part of this paper is the modeling which is conducted well. It brings new knowledge about potential habitat of the invasive fireweed.

It delivers many kinds of information in details to reader in every single aspect of the invasive fireweed so I think future reader can be satisfied with this document.

Line 163 Japan – introduced. How about Korea? Should we consider fireweed as a potential invader?

Line 333 PAs ? specify

Line 494 Integrated management: I wish to see a new figure about decision-making tree – when and on what condition to apply different management options. It would be very beneficial for land manager to decide/judge what kind of management to apply on different conditions.

Author Response

Thank you very much for the valuable suggestions. Please see the attached file.

Reviewer 2 Report

Minor remarks.

Please, unify throughout the text Senecio madagascariensis or S. madagascariensis.

Unify subtitles hierarchy style throughout the text. 

Is it still unknown (as in 1998) whether S. madagascariensis seeds can go through the digestive tract of cattle, sheep or birds and then germinate?

See some minor corrections in the attached file.

Author Response

(The authors gave the same response as above.)

Reviewer 3 Report

This seems to be a fairly comprehensive and understandable review of the invasion biology of fireweed.

I have a few concerns related largely to the likely spread of this species.

  1. The model predictions seem to be totally the opposite of the results of LeRoux, who predicted that this species might have an advantage in arid areas. Clearly the CLIMEX predictions ( 500-100mm rain) suggest regions of high precipitation and this needs some discussion/clarification. Is this a problem with the different modelling approaches, for instance, or were the previous projections in error for some reason?
  2. The CLIMEX projections are also indicative of areas where the climate might be suitable but these are very coarse and take no account of regional microclimate variations that would have to be considered. In any case these do not necessarily predict the regions where this species is likely to invade due to a range of other confounding factors, such as land use and management practices, which is actually supported by other points covered later in the manuscript. I think this needs to be clear and requires some further discussion. Too often it is assumed that the likely spread of an alien introduction is dependent on climatic factors alone whereas it is much more complicated than this. The resident community, in particular, may be important although there is little if any detailed information on the communities that are most susceptible to invasion. I would therefore recommend including a section on the characteristics/ecology of the invaded communities and to what extent they comparable/different to native communities? This may be also important for targeted management interventions.
  3. A significant deficiency is that the modelling is based on current, rather than future climatic projections, which would make the projections in this paper misleading at best. Ideally it might have been better to use a future climate scenario for this analysis.

Author Response

(The authors gave the same response as above.)
